# Production and Health Management from Grazing to Confinement Systems of Largest Dairy Bovine Farms in Azores: A Farmers’ Perspective

**DOI:** 10.3390/ani11123394

**Published:** 2021-11-27

**Authors:** Ivo Medeiros, Aitor Fernandez-Novo, Susana Astiz, João Simões

**Affiliations:** 1Veterinary and Animal Research Centre (CECAV), Department of Veterinary Sciences, School of Agricultural and Veterinary Sciences, University of Trás-os-Montes and Alto Douro (UTAD), 5000-801 Vila Real, Portugal; al62766@utad.eu; 2Department of Veterinary Medicine, School of Biomedical and Health Sciences, Universidad Europea de Madrid, C/Tajo s/n, Villaviciosa De Odón, 28670 Madrid, Spain; aitor.fernandez@universidadeuropea.es; 3Animal Reproduction Department, National Institute of Agronomic Research (INIA), Puerta De Hierro Avenue s/n, CP, 28040 Madrid, Spain; astiz.susana@inia.es

**Keywords:** herd health, milking management, production systems

## Abstract

**Simple Summary:**

This study aimed to evaluate differences and critical factors in production and health management between dairy cattle farms with fixed milk parlours (FMP), and mobile milk systems (MMS) from Azorean grasslands. According to the farmers’ perspective, calf diarrhea, calf pneumonia, infertility/poor reproductive management, and mastitis were the main problems that farms faced in 2020. FMP was associated with more advanced and mechanized production systems, with a higher adherence to preventive and biosecurity control programs, than traditional MMS farms. MMS farms also showed a greater vocation for dual-purpose farming (beef and milk), smaller herd sizes and more grazing time for cows. In conclusion, inherent and non-inherent differences in production and health management between FMP and MMS were quantified by authors. These results indicate that a greater adoption of preventive veterinary medicine and biosecurity measures should be taken, especially among MMS farms. The education of farmers should also be improved and stimulated.

**Abstract:**

The intensification of bovine milk production in the Azores has led farmers to increase farm size and specialization in grasslands, implementing confined and semi-confined production systems. Fixed milking parlours (FMP) have progressively gained more popularity, at the expense of conventional mobile milking systems (MMS). The present study aimed to evaluate the associations between production and health management in dairy cattle farms, with FMP or MMS, in grasslands (São Miguel, Azores), according to the farmers’ perspective. A total of 102 questions about production and health management were surveyed in 105 farms with >30 dairy cows each. Farms with FMP were associated (*p* ≤ 0.05) with larger herd size, better facilities, and specialized management, however, the adoption of preventive and biosecurity measures should be improved by these farmers. MMS farms implemented a lower level of disease prevention or control programs, less frequent transhumance, and showed a wider vocation to dual-purpose (milk and cross beef) than FMP farms. In conclusion, MMS and FMP farms tried to optimize yield and economic viability in different ways using grasslands. Several biosecurity and health prevention constraints were identified for improvement.

## 1. Introduction

The termination of milk quotas in the European Union has led to a more liberal and competitive market, with a more volatile and uncertain milk price [1]. As such, farms have changed their methods of production and operation. Dairy farms are increasing in size with growing efficiency, specialized work, and a higher adoption of preventive measures [2,3,4].

Areas with a mild temperature and sufficient humidity and rainfall, make pasture-based dairy production possible for bovine farms [5]. Thus, several regions worldwide, such as São Miguel Island (Azores) and New Zealand have adopted a “traditional” dairy milk production system, based on pastures with cows grazing during most of the year. Cows are naturally grazing animals, and pastures are their normal environment, where they can express normal behavior, and this theoretically creates the highest welfare level [6,7]. In fact, grazing herds achieve better animal welfare indicators, and a lower prevalence of certain diseases, such as lameness or even mastitis [8,9].

Despite this, cows fed exclusively or mostly on grazing sometimes cannot achieve their nutritional needs, and are thus limited in their productivity [10]. Moreover, one of the key limitations of a full grazing system has historically been hand milking. The first milking system was introduced in the 1800s, and has greatly evolved up to the present day [11], with the surge of robotic milking systems. Herringbone parlors were invented in New Zealand (1952), featuring advances in vacuum systems [12], to provide milking rooms for large-scale milking. Fixed milking parlors (FMP) are also used in grazing systems, but mobile milking systems (MMS) have also been developed, thus allowing the use of separate or rotational grasslands [13]. Other advantages are keeping cows in their natural environment, independently of distance from farming facilities, and reducing stress due to excessive animal movement.

The fall of the price of milk and an increasingly competitive market have led to the need to increase farm efficiency [14]. In the case of improvement of farms on pastures, farmers have chosen different production alternatives by using more intensified management [15]. Furthermore, some farms have reduced or removed all grazing, switching to an indoor-based system. Simultaneously, efforts have been made to optimize outdoor systems [16], including the use of grazing-based robotic milking systems, and improving the quality of grass, grass growth, grazing conditions, and grazing management [17,18,19].

There is an obvious relationship between veterinary health management and types of dairy systems, with health management progressively changing from curative to preventive programs with increasing farm size and individual milk yield [3]. The increases in milk production and farm size can lead to negative effects on cows, including the rising prevalence of certain diseases, or the emergence of new health issues that are directly linked to this kind of system [20], which are related to (reduced) cattle resilience [21]. These kinds of problems can be observed, in both FMP and MMS systems, with a certain level of intensification of their management systems [22].

Traditionally, MMS are the most often used milking systems in the Azores [5], based on 100% grazing time and dairy cattle transhumance between grasslands. In recent decades, together with general management and production system changes, these milking systems have been progressively replaced by FMP. Although these changes are occurring, there have been few formal studies on the level and impact of this shift in this region. However, it is known that the effects can be diverse [23], and that there are areas in the world, such as New Zealand and Ireland, where although the systems have evolved, the usage of pastures has prevailed [24,25]. Despite their relevance, the impacts of these changes on farm performance, management conditions, and animal health of these “changing” or “staying traditional” systems in the Azores, have been ignored.

Therefore, the aim of this study was to recover information on how Azorean farms are actually functioning in terms of production and health management. This was done by surveying the main actors and drivers of these changes, i.e., the farmers of Azorean dairy cattle farms on São Miguel Island, and differentiating between farms using MMS, which are “traditional systems”, or using FMP, which are “evolved systems”. Our hypothesis was that health management shows several differences depending on the production system, due to inherent differences that can be quantified and described by farmers. The ultimate goal of this study was to characterize herd health management of these dairy farms, thus identifying critical factors that should be improved, in order to increase Azorean dairy industry competitiveness.

## 2. Materials and Methods

### 2.1. Local, Sample Size and Selection of Farms

The survey addressed dairy farmers of São Miguel Island (Azores). Located in the middle of the Atlantic Ocean, the Azores is an archipelago with 9 islands, with mild temperatures (minimum and maximum temperature: 12.0–18.4 °C, respectively), humidity (minimum and maximal relative humidity: 89.0–97.4%) [26,27], and an abundant rainfall climate with precipitation of 960.6 ± 201 mm per year, with 75% of the precipitation falling between October and March [26,27].

Considering the mean size of Azorean farms (26 ± 3.9 cows; ±SD), only dairy and dual-purpose (from crossbreeding calves) farms with ≥30 adult cows were selected to obtain the largest ones due to their potential economies of scale. A total of 110 questionnaires were personally distributed by the first author from February to April 2021. Farmers were interviewed face to face (15 interviews were performed by the first author), or online via Google Forms [28] (90 respondents), after having personally contacted the farmer or having talked to them over the telephone.

Intensive or semi-intensive production systems were defined according to the time that the cows stayed indoors (confined), with availability of a milking parlor or fixed fixed milking parlors (FMP), or outdoors (grazing) using mobile milking systems (MMS), respectively, during the whole or greater part of the day. Cows from semi-intensive production systems grazed for at least 8 h per day [29]. All the farms used cattle with a genetic merit for milk-yield production.

### 2.2. Survey

The questionnaire addressed herd health management issues and major health problems observed during 2020 by farmers, as a modified version of the questionnaire used in a previous study [30]. Our questionnaire was completed by a preliminary assessment of management practices occurring in Azorean farms. The structured questionnaire consisted of eleven topics: characterization of the farm; biosecurity; calving and fertility; rearing management (up to 12 months of age); lameness; nutrition; reproduction; milking practices and mastitis; disease prevention; dry cow management; and major problems.

The questionnaires consisted of 102 questions, both closed and binary, with one additional table to express ordinal categories of intensity of the referred problem/issue. A total of 15 questions addressed management indicators and farm characteristics. A mastitis prevalence 5-point scale was proposed for affected cows: 1: 10%; 2: 10–20%; 3: 20–30%; 4: 30–40%; and, 5: >40% of prevalence. For health problem intensity estimation, the following scale was proposed: 1-Not problematic; 2-Less problematic; 3-Problematic; 4-Quite Problematic; 5-Serious/uncontrolled at the farm level.

### 2.3. Statistical Analysis

All data were recorded and statistically analyzed under the regulations of the General Data Protection Regulation (GDPR), in accordance with European regulation [31].

The minimum sample size of questionnaires (*n* = 99) was calculated according to Thrusfield, by considering the 95% confidence level, a 5% margin of error (Z = 1.96) and a 93% expected response rate, and adjusted for finite populations (3247 farms) [32].

All information was coded numerically, in order to assist analysis and guarantee anonymity. Uncategorized data were recoded into ordinal level data. Categorical data were entered into a database. Surveys not fully answered contributed partially to the responding topics, with a univariate model applied to maximize the number of answers per question. Two groups were formed to compare differences, depending on the type of milking systems used: FMP versus MMS farmers (Figure 1). Differences between percentages were evaluated with the Pearson chi-square test. A non-normal distribution of all continuous variables, including all five-point scales from categories of intensity/prevalence, was confirmed using the Shapiro–Wilk test. Therefore, a non-parametric one–way ANOVA model, followed by Van der Waerden post hoc analysis to test significance, was used [33]. The results are described as the mean percentage, or mean score and variation, as ±SEM for a significance level of 0.05. JMP^®^ 14 software for Windows (SAS Institute, Cary, NC, USA) was used.

## 3. Results

The overall response rate was 95.5% (105/110). In the respondents, we found 82.9% (87/105) with FMP, versus 17.1% (18/105) with MMS.

### 3.1. Characterization of the Farms, Biosecurity and Veterinary Advice

The estimated farm size (total cows) was 35% higher in FMP (213.1 ± 11.7; *n* = 87) than MMS (157.8 ± 21.9; *n* = 18; *p* < 0.05; Appendix A) farms. A higher proportion of dual-purpose farms Artificial Insemination (AI) performed with beef breeds to have crossbreed calves), was observed in the MMS (50.0%; 9/18) as opposed to in the FMP group (24.1%; 21/87; *p* < 0.05). Semi-intensive dairy production was observed in 74.3% (78/105; *p* < 0.001) of the farms without differences between groups (*p* = 0.33). The remaining farms practiced intensive production.

Herringbone parlor (64.4%; 56/87), tandem parlor (34.5%; 30/87) and robotic milking machines (1.1%; 1/87) were observed in FMP farms. All MMS farms had Herringbone parlors. Refrigerated milk bulk tanks were more frequent in FMP (90.8%; 79/87) than in MMS farms (22.2%; 4/18; *p* < 0.001), as was the inclusion into an official animal welfare program (44.8%; 39/87 and 16.7%; 3/18, respectively, *p* < 0.05).

A higher proportion of FMP farms (58.6%; 50/87) had isolated sick pens/bays for sick animals than in MMS farms (27.8%; 5/18; *p* < 0.05). Animal transhumance is a common practice in the Azores, and it is defined as the movement of animals using public roads, so the animals can move from one pasture to another. This practice was less frequently practiced in FMP (42.5%; 37/87) compared with MMS (83.3%; 15/18; *p* < 0.01) farms.

The veterinary assistance provided by veterinarians working exclusively with farmer co-operatives, partially nationally subsidized, differed between types of farms (MMS: 94.4%; 17/18; FMP: 57.1%; 48/84; *p* < 0.01), with 21.8% (19/84) of FMP farms reporting veterinary assistance only from private veterinarians (21 farmers did not respond to this question).

### 3.2. Reproductive Management

No differences in breeding methods were found between farms (Table 1), with artificial insemination being mainly and almost exclusively implemented in adult cows, while in the heifers, the choice fell mostly to natural mating. The estimated mean number of services per pregnant (P) adult cow was very similar between FMP (2.3 ± 0.2 AI/P) and MMS (2.1 ± 0.1 AI/P; *p* = 0.99) farms.

Reproductive management included reproductive examination up to pregnancy (*p* = 0.002), estrus or ovulation induction/synchronization protocols (*p* < 0.001) and routine pregnancy diagnosis (*p* = 0.002), which was more frequently implemented in FMP farms.

In FMP farms, most of the abortions were in the middle of gestation (3–6 months of pregnancy), while in MMS farms, the majority of abortion tended to occur up to six months into gestation (*p* < 0.01). The laboratory diagnosis of abortive agents at abortion occurrence during 2020 was low (10.5%; 15/105), with no significant differences (*p* = 0.49) between types of farms. Nevertheless, and according to the history of the farms, in 76.2% (32/42) of the cases, an infectious/toxic etiology was identified in FMP farms (*p* < 0.001).

### 3.3. Rearing Management (Up to 12 Months)

Seasonal calving distribution, according to grass availability, tended to be more frequent in MMS farms (22.2%; 4/18; *p* = 0.07; Appendix A). Calving pens were more common in FMP farms (51.7%; 45/87; *p* < 0.01). Retained placenta was classified as a significant problem for 25.7% (27/105) of the farmers, with no significant differences between farms (*p* = 0.83). FMP farms mainly (64%; 55/86) buried placentas at pasture, while MMS farms did not discharge them out at all (44.4%; 8/18; *p* < 0.05).

Appropriate calf barns were also more commonly observed in FMP (89.7%; 78/87) farms compared with MMS (72.2%; 8/13; *p* = 0.05) farms. Colostrum tended to be administered for more days in MMS (4.7 ± 0.3 days) than in FMP (3.9 ± 0.5 days; *p* = 0.10) farms. Most farms, independent of milking system, did not store colostrum (15.2%; 16/105; *p* = 0.45) but considered diarrhea as their main problem at calving (69.5%; 73/105). Pneumonia in calves was considered a major problem in MMS farms (72.2%; 13/18; *p* < 0.01), which also reared males calves more frequently for fattening (61.1%;11/18; *p* < 0.01). Preventive measures, such as vaccination of pregnant cattle to prevent pneumonia and diarrhea in calves (25.3%; 22/87 vs. 5.6%; 1/18, respectively; *p* = 0.07) and vaccination of calves up to 12 weeks of age (13.8%; 12/87 vs. 0.0%; 0/18, respectively; *p* = 0.09), tended to be more frequently adopted by FMP than MMS farms.

### 3.4. Nutrition and Metabolic Disease Prevention

A higher level of nutritional assessment was performed in FMP farms. Nevertheless, both farm groups were regularly assessed by a nutritionist (88.6%; 93/105; *p* = 0.96), regardless of the type of farm. Body condition scoring was more commonly implemented in FMP farms (60.9%; 53/87; *p* < 0.05; Table 2).

Forage nutritional analyses (*p* < 0.001), as well as diet adjustments (*p* < 0.01), were more frequent in FMP farms than in MMS farms. Additionally, the total mixed ration (TMR) system, also named the “unifeed” system, was more frequently found in FMP farms (*p* < 0.001), with forages coming from their own production in all MMS farms (*p* = 0.02).

Water was administered ad libitum in all farms. Mostly, the water of both groups of farms came from the municipal water supply (72.4%; 76/105).

### 3.5. Milking and Mastitis

Differences were observed in the milking procedures of farms. Pre-dipping (*p* < 0.001) and paper towels (*p* < 0.001) but not post-dipping (*p* = 0.30) were more frequently implemented in FMP farms (Table 3). 

Mastitis incidence and mortality did not differ among farms (18.1%; 19/105; *p* = 0.62), but the estimated somatic cell count tended to be higher in MMS farms (*p* = 0.07).

### 3.6. Lameness

Lameness was considered a major problem in 41% (43/105) of all farms. Nevertheless, only 34% (33/105) of farmers implemented a continuous lameness control program (Appendix A). Approximately half (48.3%; 42/87) of the FMP farms had a functional footbath, contrary to MMS farms (5.6%; 1/18; *p* = 0.001). On most of the farms (77.2%; 78/101), trimming was only performed as treatment after the detection of lameness, without differences between groups.

### 3.7. Drying-Off and Prepartum Care

No significant differences in cow management between groups were observed at dry off, including the length of the dry period (45–60 d; Appendix A). Drying-off anti-biotherapy was administered in 94.3% (99/105) of the farms. The mean percentages of the main procedures during the dry period and prepartum are reported in Figure 2.

The dry cows joined lactating cows in the prepartum period in 90.5% (95/105) of the farms without differences between groups (*p* = 0.26). FMP farms (95.3%; 82/86) tended to include a higher percentage of pregnant heifers in the lactating herd during the prepartum period, compared with MMS farms (83.3%; 15/18; *p* = 0.06).

### 3.8. Disease Prevention and Major Problems

In general, a low to moderate level of preventive measures was implemented in farms, with some differences between FMP and MMS groups (Table 4). The frequency of blood sampling for disease diagnosis was higher in MMS (11.1%; 2/18) than in FMP (1.2%; 1/87; *p* < 0.05) farms. Conversely, the use of insecticide during the hot season/periods was more frequent in FMP farms (86.2%; 75/87; *p* = 0.05).

Vaccination for clostridial diseases, bovine rhinotracheitis virus/bovine viral diarrhea, mastitis and/or respiratory complex disease was implemented in 42.9% (45/105) of the farms, with no differences between FMP and MMS (*p* = 0.50) groups.

Blood sampling for mineral quantification and fecal sampling for parasitic disease diagnosis were not performed by any of the farms questioned.

According to the farmers surveyed, no natal differences were observed regarding the problems that affected FMP and MMS dairy farms in 2020 (Figure 3). However, neonatal diarrhea (score point = 4.0 ± 0.2 vs. 3.6 ± 0.1; *p* = 0.04) and infertility/reproductive problems (score point = 3.3 ± 0.2 vs. 2.9 ± 0.1; *p* = 0.05) were slightly more problematic for MMS farms, as opposed to FMP farms.

Neonatal diarrhea, calf pneumonia, mastitis, retained placenta, metritis, lameness and infertility/reproductive problems were the most problematic diseases/issues indicated by farmers from both groups of farms. 

## 4. Discussion

Overall, larger herds, better facilities, greater focus on prevention, and constant nutritional and reproductive assessments were more frequently observed in FMP farms than in MMS farms. In contrast, a higher proportion of dual-purpose farms (milk and beef production) was found among the MMS farms trying to increase income. Therefore, the single purpose to produce milk, observed more often in Azorean FMP farms than in MMS farms, seems to be in line with the new reality of the European market, which has been without milk quotas since 2015, requiring a more efficient and specialized milk production.

The (semi)intensification level of dairy milk production systems (>70%), was similar in both farm groups in our study. In fact, supplementation to grazing dairy animals is required to maintain an adequate level of milk yield [34], together with an improvement in the stocking rate in the Azores, e.g., 3.2 cows/ha, as reported by Morais et al. (2018) [35]. Despite this last issue, important management differences were observed between the types of farms.

Regarding reproductive management, differences were observed between the types of farmers surveyed (see Table 1), with FMP farmers more frequently implementing complete reproductive management protocols and appropriate tools. This is to say, reproductive examination during open days, to select cows for breeding, and to treat diagnosed pathologies, induction and synchronization of estrus and ovulation, pregnancy diagnosis, and ultrasonography. All these interventions are crucial to optimize the reproductive output of dairy farms [4]. Further research is required to quantify differences between production systems.

In our study, a total of 39% of the farmers, independent of farm group, used at least one ancillary device to detect estrous. This result denotes the progressive adoption of technologies to improve fertility. In a similar survey of Canadian dairy farms [36], 89% of farmers used visual detection of estrous as their only method (3.5 observation times per day), i.e., only a low proportion of farms used ancillary devices. Nevertheless, in this last study, fixed-time artificial insemination was mainly implemented [36]. However, it is well known that several devices to detect estrus have been largely and efficiently implemented worldwide [37,38].

Failure in heat detection and low conception rates are major reproductive problems [39], with infertility and poor production being the main causes for culling dairy cows in the USA [40]. In fact, poor reproductive management represents losses of up to 231€ per cow per year, due to a decrease in milk yield and a high calving interval [39]. Therefore, the implementation of adequate reproductive strategies [41] is essential in dairy farms and should be enhanced in Azorean farms, especially among MMS farms. Pregnancy diagnosis (*p* = 0.01), protocols for estrus/ovulation induction (*p* < 0.001) and reproductive examination during open days (*p* = 0.002), were performed less often in MMS farms than in FMP farms. All these results were expected, since indoor systems allow better reproductive control and management [42].

In the present study, the number of calves born on the farms (2020) was, as expected, higher in FMP farms, since these farms had larger herd sizes (213.1 ± 11.7 and 157.8 ± 21.9 total animals for FMP and MMS, respectively; *p* < 0.05), and the trade of pregnant heifers or adult cows remained low. Appropriate calving pens were more commonly found in FMP farms, which are essential to control the vitality of the newborns, adequate immunity transference, and the mothers’ health [43]. Nevertheless, the advantages and limitations of calving indoors or outdoors are still up for debate [44].

In MMS farms, we observed a higher tendency towards seasonal calving according to grass availability (*p* = 0.07). This is due to the importance of grass in the production system of these farms. This practice is in line with what is very common in Ireland and New Zealand, where seasonal calving is largely adopted, so local farms can take advantage of their animal production potential at the time of grass growing [45]. Additionally, in our study, only 30% of farms minimized calving during summer. This calving seasonality has the potential advantage of mitigating calving heat stress [46], and initiating new lactation in more developed grass periods, even if forages are stored. It has been observed that calving in late winter is most profitable in grazing systems, independent of milk premium price [47].

Rearing the animals is a very sensitive part of the production system, with correct hygiene, general management, and appropriate colostrum administration being essential to prevent the main health problems for young cattle: enteric and respiratory diseases being responsible for the highest mortality and morbidity rates [48,49,50].

It is essential to provide a sufficient volume of high-quality colostrum in the first hours of life [51]. Very few farms in our study (17.2% FMP and 11.1% MMS farms) had colostrum storage banks, and even fewer had an appropriate instrument to evaluate the quality of the colostrum prior to storage. All these aspects, as well as cleanliness of the pens and colostrum quality assessment, have been associated with pneumonia or diarrhea in calves [52]. Therefore, if these practices improved, the incidence of neonatal diseases would possibly decrease on São Miguel dairy farms (see Figure 2). Indeed, diarrhea and pneumonia in calves were the main problems faced by Azorean farmers, which is another issue to be addressed. In addition to direct losses [49], it is known that calves raised without diarrhea and/or pneumonia achieve greater longevity, yield, and profitability; furthermore, they are healthier cows [53]. In our study, FMP farms more frequently had calf sheds, administered colostrum for more days, and utilized more preventive measures in rearing than MMS farms did, evidencing poorer rearing management on these MMS farms.

Additionally, a tendency of FMP farms to use vaccination protocols more frequently, to prevent pneumonia and diarrhea in calves, was observed (see Appendix A). Vaccination of dams and calves up to 12 months probably contributed to approximately half of the FPM farmers considering pneumonia as the main problem in their calves (34.5%; 30/87) when compared to that reported by MMS farmers (72.2%; 13/18; *p* < 0.01). Nevertheless, calf density can also contribute to pneumonia in MMS, and is higher in that type of farm due to its dual-purpose production. Moreover, calves exposed to extreme weather may not be able to regulate their body temperature with their own thermoregulation mechanisms, leading to significant losses in performance, and average daily gain [54]. This circumstance is more frequently observed in MMS farms than in FPM farms.

In terms of biosecurity, FMP farms adopted more measures, such as the existence of spaces to isolate sick animals, and less movement of animals using public roads. However, biosecurity measures were scarcely implemented by the dairy farms surveyed on São Miguel Island, which is an important observation of the study. In MMS farms, there was a tendency for more frequent movement of animals (transhumance), and of purchases and sales of living animals (*p* = 0.10). Despite this, only 38.5% of the respondent farmers (see Appendix A) implemented quarantine measures when introducing animals. The movement of animals is the main cause of the spread of disease, and the introduction of new animals is a farm’s highest risk of being infected with new diseases [55,56]. Moreover, important biosecurity practices, such as providing their own clothing to visitors or the use of footbaths, were not implemented at all in the farms surveyed on São Miguel Island. For example, in Belgium, between 66% and 61% carry out such methods [55]. The different kinds of production systems found in the Azores probably contribute to these differences. Finally, all Azorean farms bury dead animals on pastures, a practice almost abandoned in the rest of the world. In the USA, only 27.3% of farms bury their dead cows, while 29.2% add carcasses to compost, and 27.2% put them to render [40]. In fact, the burial of dead animals in the EU is forbidden, preventing contamination of soils and water by pathogenic and zoonotic microbials [57]. Nevertheless, some exceptions [58] are made in remote areas and specific conditions, such as Azores.

More frequent forage nutritional analysis, mainly when a new diet is available, and diet adjustment, according to lactational curve of dairy cattle, should be improved by farmers to optimize milk yield [59]. Cows on FMP farms spent more time indoors, making it easier to control the diets and intake rate, and to provide adequate feed quality, as demonstrated in previous studies [10]. Adequate nutritional management is essential for cattle health, welfare, reproductive performance, and milk yield [42], thus also being essential for farm productivity and economic sustainability [42,60]. In the present study, MMS farms, when compared to FMP farms, received nutritional assessments less frequently (see Table 2). As a result, fewer forage nutritional analyses were conducted (*p* < 0.001), and fewer scoring body conditions were performed (*p* = 0.002) on MMS farms. Consistent with this, the TMR system was more commonly observed in FMP farms (*p* < 0.001), with this technology allowing better nutritional management, and the possibility of giving animals homogeneous feed and a balanced diet [10,61,62].

Nutrition is also fundamental at the peri- and postpartum periods, during which cows undergo several metabolic and hormonal changes, which compromise their immune functions [63]. Negative energy balance is common in postpartum dairy cows, due to low feed intake and the inability to reach cow energetic requirements for milk yield [64]. Appropriate nutritional management of the peripartum cow is essential to avoid such problems [65], together with other practices [66], reducing the incidence of postpartum problems such as retained placenta, mastitis, metritis, and endometritis. These issues can all lead to several economic losses, and early culling rates [66,67]. Certain nutritional supplements, such as vitamin D, lysine, methionine, and others can be considered, in order to increase performance after calving [68,69]. In our study, monensin supplementation was the most common practice (27.6%).

In addition to the type of machine used to milk (used to classify the type of farms), relevant differences were found in the milking routines and hygiene practices during milking (see Table 3), with FMP farms implementing indicated routines more frequently [70], such as pre- and post-dipping and the use of paper towels. It is easier to implement adequate milking routines in fixed milking parlors than with mobile milking systems [71,72]. However, even when observing only FMP farms, we detected a huge difference in the possibilities of receiving specialized technical advice on the island, compared to that of other regions in Europe and Portugal; this clearly limits the implementation of new technologies, and the possibility of adequate evolution in the Azores [5], despite it being one of the main milk-producing regions in Portugal. Adequate milking practices, new technologies, and udder health control programs are key points to control mastitis. Although mastitis was not one of the top three most important diseases affecting São Miguel dairy farms, according to farmers’ perceptions, it induces large economic losses which is in in agreement with the literature [73,74,75].

After rearing, reproductive problems, mastitis, and placenta retention were the challenges highlighted by our farmers as their major problems (see Figure 3). Similarly, infertility, clinical mastitis and lameness were the principal diseases that affected American dairy farms in 2014 [40].

In our study, 41% of the farmers considered lameness as a major problem, which can be considered a high prevalence for grazing systems. There are several causes that can justify this evidence, for example, the influence of (hard) floors found in public roads (transhumance); the high, steady humidity, mainly in winter season; scarcity of trimming programs and footbath use. Lameness negatively influences productive and reproductive traits [76], causing significant economic impact to farms [77].

There is a low to moderate adherence/application of preventive measures to combat these and other diseases on São Miguel Island. Only 42.9% of the surveyed farms carry out at least one vaccination. There was a large discrepancy when compared to data from previous studies, such as on Irish dairy farms, where only 13% did not apply any vaccinations [78]. The low vaccination rate observed in our study probably relates to the cost of vaccination, lack of immediate health improvement, previous experience of failure to control the disease by the farmers, and a general lack of education of farmers. However, vaccination by itself is not enough. Proactive co-operation between veterinarians and farmers is essential to optimize health and sustainability in dairy farms [79].

All these needs detected in the present study indicate an urgent necessity for improvement in farmer education, as previously seen in other regions [49]. Assuring that the farmers have the knowledge and equipment to record their own data, and are then able to calculate and interpret basic indicators, is essential to the early detection of diseases and problems [2]. In fact, the greater part of the health information reported in this study came from farmers’ perception. A health data record is important to objectively evaluate the herd health and take appropriate decisions, and can contribute to a low degree of health program implementation. Nevertheless, farmer education is only a part of the equation to drive behavioral change. According to Michie et al. [80] individual behavior changes are related to capability (knowledge and skills, via education), motivation (brain stimulating process) and opportunity (outside events). The dairy industry is a business which provides incomes to farmers, and is the strongest motivation for farmers. The production efficiency and herd health management improvements, and new opportunities coming from social demands, including consumer perspectives on animal welfare and environmental impact, seem to be crucial keys for behavioral changes.

Consumers have a general opinion that grazing cattle are in a better condition in terms of health and welfare [81]. This is important and may change the markets. In fact, consumers have increasingly higher interest in animal production conditions, animal welfare, sustainability of the systems, and environmental protection [82], which are the main drivers of changes in legislation, especially in the European Union [83]. All of these drivers of change can be considered an opportunity for areas that produce mainly based on pasture.

This evolution also must be compatible with preserving the particular aspects of the Azorean Islands, and production systems closely linked to the natural environment and to agrarian populations [5,81,82]. The environmental impact of the dairy industry should be attenuated, ensuring economic viability of the farms with dairy added-value programs such as “happy cows” [5].

Our study reveals that both FMP and MMS have the potential for progress, improving their efficiency and preserving animal welfare, in extensive and intensive production systems.

## 5. Conclusions

There is a clear difference in health management between these two types of farms; MMS farms are associated with a more traditional production approach, while FMP farms have transitioned to more specialized dairy systems. However, all farms on São Miguel Island would benefit from animal welfare, productivity, resource efficiency, and sustainability. This would be further improved by the implementation of preventive and structured control programs, assessed by professional advisors, and thus enhance health, welfare efficiency and profitability. However, São Miguel Island dairy farms must continue taking advantage of their idiosyncrasy, benefiting from the natural resources available, stressing the production of “green milk”.

## Figures and Tables

**Figure 1 animals-11-03394-f001:**
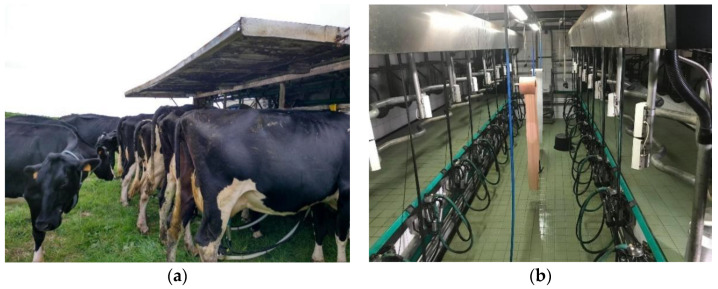
Traditional mobile milking system (**a**) and fixed milking parlors (**b**).

**Figure 2 animals-11-03394-f002:**
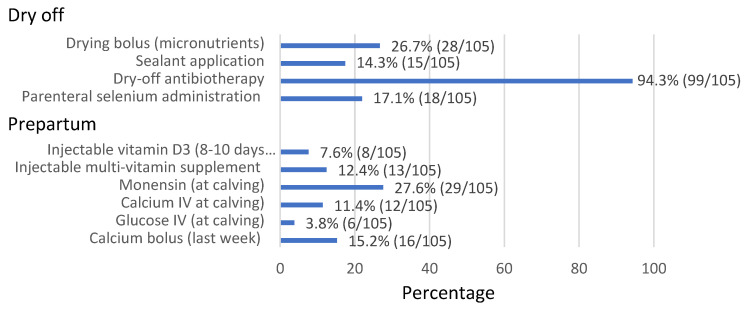
Main practices, micronutrient, and calcium administrations at dry-off and prepartum periods.

**Figure 3 animals-11-03394-f003:**
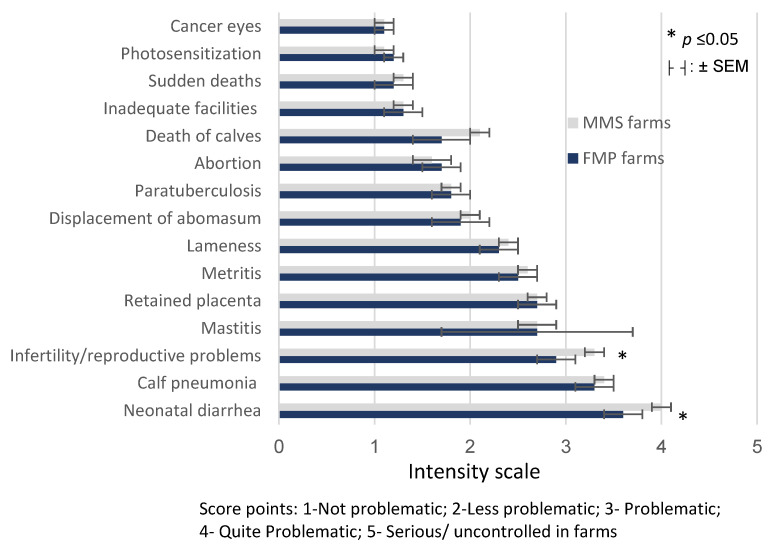
Score intensity scale of major problems suffered in 2020, according to the farmers’ perceptions, in dairy farms with mobile (MMS) versus fixed (FMP) milking systems.

**Table 1 animals-11-03394-t001:** Reproduction management in dairy farms with fixed (FMP) and mobile (MMS) milking systems.

Factor	Farms		*p* Value
	FMP	MMS	
Breeding			
Adult females (number of animals)	139.1 ± 8.0 (32–400) ^(1)^	100.8 ± 13.0 (20–200)	0.04
Heifers (number of animals)	31.5 ± 2.3 (8–130)	24.8 ± 3.9 (8–70)	0.17
Age at first breeding (months)	16.1 ± 0.3 (12–20)	15.3 ± 0.7 (12–18)	0.14
Breeding methods—Adult cows			
Artificial insemination (*n* = 46)	42.5% (37/87) ^a (2)^	50.0% (9/18) ^a^	0.82
Natural service (*n* = 8)	8.1% (7/87) ^b^	5.6% (1/18) ^b^	
Both (*n* = 51)	49.4% (43/87) ^a^	44.4% (8/18) ^a^	
Breeding methods Heifers:			
Artificial insemination (*n* = 18)	16.1%(14/87) ^a^	22.2% (4/18) ^a^	0.82
Natural service (*n* = 75)	72.4% (63/87) ^b^	66.7% (12/18) ^b^	
Both (*n* = 12)	11.5% (10/87) ^a^	11.1% (2/18) ^a^	
Artificial insemination performed by (*n* = 103):			
Technician (*n* = 94)	91.9% (79/86) ^a^	88.2% (15/18) ^a^	0.08
Farmer (*n* = 8)	8.1% (7/86) ^b^	5.9% (1/18) ^b^	
Both (*n* = 1)	0.0% (0/86)	5.9% (1/18) ^b^	
Reproductive management:			
Beef sire semen (*n* = 103) ^(4)^	92.9% (79/85)	94.4% (17/18)	0.82
Estimated mean number of services per pregnancy, cows (*n* = 103)	2.3 ± 0.2 (1–5)	2.1 ± 0.1 (1–4)	0.99
Reproductive examination during open days ^(3)^ (*n* = 104)	73.6% (64/87)	35.3% (6/17)	0.002
Ancillary oestrus detection devices	39.1% (34/87)	38.9% (7/18)	0.99
Protocols of oestrus or ovulation induction/synchronization	65.5% (57/87)	16.7% (3/18)	<0.001
Pregnancy diagnosis	79.3% (69/87)	50.0% (9/18)	0.01
Pregnancy diagnosis method (*n* = 78):			0.002 0.002
Manual transrectal palpation (exclusively)	13.0% (9/69) ^a^	55.6% (5/9) ^a^	
Ultrasonography	87.0% (60/69) ^b^	44.4% (4/9) ^b^	
Abortion: Estimated total abortion number detected by farmer in 2020	3.5 ± 0.3 (0–10)	3.1 ± 0.5 (0–8)	0.41
Abortion timing (*n* = 101):			
Up to 3 months	10.7% (10/84) ^a^	41.2%(7/17) ^a^	0.005
3–6 months	69.1%(58/84) ^b^	52.9% (9/17) ^b^	
>6 months	17.8% (17/84)% ^a^	5.9% (1/17) ^a^	
Venereal disease diagnosis of sires (mating; *n* = 57)	2.3% (1/43)	8.3% (1/12)	0.33
Laboratory diagnosis, according to farm history, of (*n* = 42):			
IBRV (*n* = 16)	75.0% (12/16)	25.0% (4/16)	0.005
BVDV (*n* = 16)	81.3% (13/16)	18.8% (3/16)	<0.001
Neosporosis (*n* = 8)	75.0% (6/8)	25.0% (2/8)	-
Toxin/fungi (*n* = 2)	50.0% (1/2)	50.0% (1/2)	-

^a, b^ Different superscript letters for the same column: *p* < 0.01. %: Percentage of farms with an affirmative response; *n*: Number of respondents. Omitted values means *n* = 105. ^(1)^ arithmetic mean ± standard error of mean (min–max). ^(2)^ (*n*/N): number of affirmative responses/number of total respondents. ^(3)^ Previous evaluation of uterine involution/content and ovarian examinations of breeding cows. Abbreviations: BVDV, bovine viral diarrhea virus; IBRV, infectious bovine rhinotracheitis virus. ^(4)^ Beef sire semen was used in selected dairy cows for crossbreeding purposes to obtain beef calves.

**Table 2 animals-11-03394-t002:** Nutritional management and metabolic disease prevention in dairy farms with fixed (FMP) or mobile (MMS) milking systems.

Factor	Farms		*p* Value
	FMP	MMS	
Nutritional assessment: Scoring body condition (peripartum; *n* = 104)	60.9% (53/87) ^(1)^	29.4% (5/17)	0.02
Forage nutritional analyses (FNA, *n* = 104)	93.1% (81/87)	58.8% (10/17)	<0.001
Diet adjustment based on FNA results (*n* = 94)	90.5% (76/84)	60.0% (6/10)	0.006
Feeding management:			
Unifeed system (*n* = 104)	80.5% (70/87)	41.2% (7/17)	<0.001
Adding concentrate feed to unifeed (*n* = 75)	50.0% (34/68)	42.9% (3/7)	0.72
Own forages	93.1% (81/87)	100% (18/18)	0.25
Corn silage ^(2)^	94.3% (82/87)	88.9% (16/18)	0.41
Grass silage (*n* = 100) ^(3)^	8.3% (7/84)	6.3% (1/16)	0.78
Baled grass silage (*n* = 103) ^(4)^	98.8% (83/84)	100% (17/17)	0.66
Hay rolls (*n* = 101) ^(5)^	23.5% (20/85)	25.0% (4/16)	0.90
Straw (*n* = 103) ^(6)^	32.6% (28/86)	43.8% (17/16)	0.39
Access to pasture (grassland)	73.6% (64/87)	94.4% (17/18)	0.06
Feed concentrate during milking	85.1% (74/87)	100% (18/18)	0.08
Dry cow diet ^(7)^ (*n* = 104)	23.0% (28/87)	58.8% (10/17)	0.003
Feed concentrate to dry cows	32.2% (20/87)	22.2% (4/18)	0.40
Water source of the farm:			
Pit water (*n* = 2)	1.1%^a^ (1/87) ^a^	5.6% (1/18) ^a^	0.32
Riverside (*n* = 25)	25.3% (22/87) ^b^	16.7% (1/18) ^a^	
Municipal water supply (*n* = 76)	72.4% (63/87) ^c^	72.2% (13/18) ^b^	
Wellspring (*n* = 11)	1.1% (1/87) ^a^	5.6% (1/18) ^a^	

*n*: Number of respondents. Omitted values means *n* = 105. %: Percentage of farms with an affirmative response. ^a, b, c^ Different letters for the same column: *p* < 0.01. ^(1)^ (*n*/N): number of affirmative responses/number of total respondents. ^(2)^ Corn silage is used by 81.4% (79/97) of the farms during the whole year. ^(3)^ Grass silage is used by 50% (4/8) of the farms during the whole year. ^(4)^ Baled grass silage is used by 12.5% (3/24) of the farms during the whole year. ^(5)^ Hay rolls are used by 22.9% (8/35) of the farms during the whole year. ^(6)^ Only 1.4% (1/73) of farmers also fed animals alfalfa. ^(7)^ Farmers who do not use a specific dry cow diet reported that they fed cows at pasturage (*n* = 59) and/or baled grass silage (*n* = 48), corn silage (*n* = 15), straw (*n* = 6) and/or grass silage (*n* = 1) segregated from lactating cows.

**Table 3 animals-11-03394-t003:** Milking procedures and mastitis scores, according to fixed (FMP) or mobile (MMS) milking system farms.

Factor	Farms		*p* Value
	FMP	MMS	
Milking procedures			
Pre-dipping	63.1% (53/84) ^(1)^	16.7% (3/18)	<0.001
Post-dipping	98.8% (83/84)	94.4% (17/18)	0.3
Paper towels	78.6% (53/84)	11.1%(3/18)	<0.001
Gloves	47.2% (40/84)	33.3% (6/18)	0.27
Separate teatcups for mastitis cows	7.1% (6/84)	5.6% (1/18)	0.83
Teatcup disinfection after use by mastitis cows	4.8% (4/84)	0.0% (0/18)	0.35
Hot water cleaning machine	9.5% (40/84)	5.6% (1/18)	0.59
Mastitis			
Mastitis incidence (Score ^(3)^)	2.2 ± 0.1 (1–5) ^(2)^	2.1 ± 0.2 (1–5)	0.54
Culling or death of mastitic cows	17.2% (15/87)	22.2% (4/18)	0.62
Estimated somatic cells count (log 10)	2.38 ± 0.30 (2.00–2.45)	2.45 ± 0.02 (2.00–2.60)	0.07

%: Percentage of farms with an affirmative response. *n*: Number of respondents. Omitted values means *n* = 105. ^(1)^ number of affirmative responses/number of total respondents. ^(2)^ arithmetic mean ± standard error of mean (min–max). (*n*/N):. ^(3)^ Scale 1 to 5, according to the percentage of affected cows with mastitis during 2020: 1:10%; 2:10–20%; 3:20–30%; 4:30–40%; 5: >40%.

**Table 4 animals-11-03394-t004:** Preventive health measures were adopted by fixed (FMP) and mobile (MMS) milking system farms.

Factor	Farms		*p* Value
	FMP	MMS	
Preventive measures			
Disease prevalence monitoring (serum samples)	1.1% (1/87) ^(1)^	11.1% (2/18)	0.02
Parasitic disease monitoring (faecal samples)	0.0%	0.0%	-
Mineral monitoring (serum samples)	0.0%	0.0%	-
Mineral diet supplementation during dry period	62.1% (54/87)	50.0% (8/18)	0.17
Insecticide during hot season/periods	86.2% (75/87)	66.7% (12/18)	0.05
Regular deworming	56.3% (49/87)	50.0% (9/18)	0.42
Vaccination *:			
Clostridial diseases (*n* = 3)	1.1% (1/87) ^a^	11.1% (2/18)	0.02
IBR/BVD (*n* = 37)	36.8% (32/87) ^b^	27.8% (5/18)	0.47
Mastitis (*n* = 16)	13.8% (12/87) ^c^	22.2% (4/18)	0.37
Respiratory complex disease (*n* = 16)	14.9% (13/87) ^c^	16.7% (3/18)	0.90

%: Percentage of farms with an affirmative response. ^(1)^ (*n*/N): Number of affirmative responses/number of total respondents. ^a, b, c^ Different superscript letters for the same column: *p* < 0.01. *n*: number of farms with an affirmative response. Omitted values means *n* = 105. * total of farms using vacines = 61 (some farms used more than one vaccine type). Abbreviations: BVD, bovine viral diarrhea; IBRV, infectious bovine rhinotracheitis.

## Data Availability

The data that support the findings of this study are available on request from the corresponding author (J.S.).

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
