# Peer review of "Production and Health Management from Grazing to Confinement Systems of Largest Dairy Bovine Farms in Azores: A Farmers’ Perspective"

_animals, 2021, doi:10.3390/ani11123394_

Round 1

Reviewer 1 Report

A brief summary

The aim of the paper is to describe the Azorean dairy sector and identify differences in terms of health management between farms using fixed milking parlors as and farms using mobile milking systems. The article gives a very detailed, informative and comprehensive overview over a great number of factors describing Azorean dairy farms and the discussion of the result is very thorough. Moreover, it highlights strategies to further improve the Azorean dairy sector and make it more competitive as well as efficient, profitable and animal friendly.

General concept comments

The manuscript is clear, relevant for the field and presented in a well-structured manner. However, I suggest having it revised by a native speaker, as there are quite a few wordings/phrasings/spellings that did not seem correct to me (although I am not a native speaker either).

The cited references are mostly up-to-date and the list of references does not include an abnormal number of self-citations.

The manuscript is scientifically sound and the experimental design is appropriate to test the hypothesis. I was slightly confused by the categorization of farms and farmers. If the distinction is between grazing (semi-intensive) and confinement (intensive) systems (as stated in the title), why is the tested distinction one of milking systems (factor used to describe the groups)? From my point of view, this needs to be better explained.

The manuscript’s results are reproducible based on the details given in the methods section.

The figures/tables/images/schemes are mostly appropriate but could do with some re-formatting (see comments in the PDF file). The data is interpreted appropriately and consistently throughout the manuscript. Statistical analysis is straightforward and sound.

The conclusions are mainly consistent with the evidence and arguments presented, but I suggested some aspects that could/should be considered as well.

The ethics statements are appropriate and in good order. As far as I can tell there is no data availability statement.

Specific comments

Included in the PDF document.

Author Response

First, we would like to thank again the reviewers for their thorough review of the manuscript, and for their constructive feedback and pertinent comments. We have highlighted in the manuscript the changes from Referee1 in blue, yellow for Referee 2 and green for Referee 3.

Reviewer 1

A brief summary

The aim of the paper is to describe the Azorean dairy sector and identify differences in terms of health management between farms using fixed milking parlors as and farms using mobile milking systems. The article gives a very detailed, informative and comprehensive overview over a great number of factors describing Azorean dairy farms and the discussion of the result is very thorough. Moreover, it highlights strategies to further improve the Azorean dairy sector and make it more competitive as well as efficient, profitable and animal friendly.

General concept comments

The manuscript is clear, relevant for the field and presented in a well-structured manner. However, I suggest having it revised by a native speaker, as there are quite a few wordings/phrasings/spellings that did not seem correct to me (although I am not a native speaker either).

The cited references are mostly up-to-date and the list of references does not include an abnormal number of self-citations.

The manuscript is scientifically sound and the experimental design is appropriate to test the hypothesis. I was slightly confused by the categorization of farms and farmers. If the distinction is between grazing (semi-intensive) and confinement (intensive) systems (as stated in the title), why is the tested distinction one of milking systems (factor used to describe the groups)? From my point of view, this needs to be better explained.

We thank the reviewer for this comment, and apologize for the lack of clarity. In Azores, the issue of grazing vs confinement is parallel to the fact if they use mobile milking machines, or cows enter into a milking parlour. We understand that this is not obvious for other readers, and a sentence has been added into the manuscript.

Lines 122-123

The manuscript’s results are reproducible based on the details given in the methods section.

The figures/tables/images/schemes are mostly appropriate but could do with some re-formatting (see comments in the PDF file). The data is interpreted appropriately and consistently throughout the manuscript. Statistical analysis is straightforward and sound.

The conclusions are mainly consistent with the evidence and arguments presented, but I suggested some aspects that could/should be considered as well.

The ethics statements are appropriate and in good order. As far as I can tell there is no data availability statement.

We added : “ Data Availability Statement: The data that support the findings of this study are available on request from the corresponding author (J.S.).”

Lines 534-535

Specific comments

Included in the PDF document. We do specify and answer the main questions from reviewer 1, but we also had implemented every single language tip or correction she/he made. We are extremely thankful with her/his thorough review.

Language, style and format modifications.

Lines: throughout the manuscript.

L37: how was insufficiency determined?

We have rewritten the sentence: “ however, the adoption of preventive and biosecurity measures should be improved by these farmers.”

Lines 37-39

L37-38 this sentence should be revised.

We have rewritten the sentence: MMS farms implemented a lower level of disease prevention or control programs, less frequent transhumance, and showed a wider vocation to dual-purpose (milk and cross beef) than FMP farms.”

Lines 39-41

L40-41 sustainability is here mentioned for the first time; what is meant by 'using more grasslands'?

We have rewritten the sentence: “In conclusion, MMS and FMP farms tried to optimize yield and economic viability by different ways, using grasslands. Several biosecurity and health prevention constraints were identified for improvement.”

Lines 41-43

L65: possibly revise: MMS have certain benefits?

Yes, they have some other benefits. We included a sentence to enhance clarity: “Other advantages are keeping cows in their natural environment, independently of distance from farming facilities, and reducing stress due to excessive animal movement.”

Lines 68-70

L74: Remove, and start new topic.

Done.

L86: Remove and start new topic.

We moved this paragraph for the discussion section.

Lines 494-500

L93: type of dairy system?

It has been changed

Lines 79-79

L99: this sounds as if this was unexpected; yet I think it isn't.

We did rewrite the sentence in order to enhance clarity.

Line 84

L104: remove or explain the relationship in detail.

It was removed.

L112: the aim was not to analyse, but to recover information, gain knowledge etc.

It has been changed.

Line 96

L120: what exactly is meant here? Explain.

We have added a sentence to clarify: “The ultimate goal of this study was to characterize herd heath management of these dairy farms, thus identifying critical factors that should be improved, in order to increase Azorean dairy industry competitiveness.”

Lines 102-105

L130. why did you choose farms above the mean size? Explain.

We choose them to obtain the largest ones due to their potential economies of scale. A specification has been included.

Lines 115-116

L133: as a reader I would like to know how many interviews were performed and how many of the responends filled in the online survey. Done 15 plus 90 respondents for presential and online interviews

We thank the comment from the reviewer and we specify the data she/he suggested.

Lines 118-119

L134: add proper citation.

As well, we included the reference.

Line 119

L 140: Please provide comprehensive definitions of all systems

We included a reference.

Line 125

L174: this is confusing; it is not the response rate of the FMP/MMS farmers but their percentage of all farmers.

We thank the suggestion and improved the readability. Changes have been included in the manuscript.

Lines 163-164

L178: how was dual-purpose defined?

It was referred to the breed which they performed the AI in order to have crossbreed calves. Moreover, we include a specification, although we defined it previously.

Line 168

L183-184: were the MMS always the same or were there differences too?

All MMS farms had Herringbone parlours. A specification was included to enhance clarity.

Lines 173-174

Ln 188. Animal transhumance definition.

“Animal transhumance is a common practice in the Azores, and it is defined as the movement of animals using public roads, so the animals can move from one pasture to another.”

Lines 178-180

L192: how about the remaining 17 farms?

These farmers did not answer to this question. We specified it in the revised version of the manuscript.

Lines 185-186

L200, Table 1: please revise the formatting; consider bold/italic format instead of underscore and/or indentation of variables (as in table 3).

Done throughout all the manuscript and in all tables of the supplementary file

Ln200. “When is beef semen used?”

Beef sires’ semen was used in selected dairy cows for crossbreeding purposes to obtain beef calves. A clarification has been included.

Line 200

Ln200. Estimated total abortion number: is this the total number of abortions? please give a comparable unit, e.g, no of abortions/no of calvings per year. Abortion timing (n=101): please explain this variable; is this when most abortions take place?

We added “detected by farmer in 2020” in  table 1.

Ln 200. how come total numbers per milking system are equal? they each add up to 41.

We are sorry for the mistake. It has been corrected table 1.

L226, 231: as opposed to what number of days in MMS farms?

It has been corrected.

Lines 218-220 and 226-227

L321-333: explain, what exactly is meant here? How does this aspect relate to the aspects mentioned in the first sentence?

We rewrote the paragraph to enhance clarity.

Lines 321-326

L336: This is a hypothesis, but did MMS farms really show poorer reproductive performance? Do you have data on that or only farmers' perceptions as in Fig. 2?

We do not, It is farmers’ perception. As further research is required to quantify differences between production systems, we include it in the manuscript.

Lines 339-340

L340-343: Is this really relevant, does the Canadian sample really compare to the Azorean sample? This is a very general thing to say (no reference) and it opposes what you wrote earlier, that dairy systems (especially FMP farms) are evolving rapidly.

We thank the comment for the reviewer and we rewrote the entire paragraph to enhance clarity.

Lines 342-348

L335: I suggest to remove this sentence or at least move it.

Removed.

L361: I would expect higher vitality if calving pens are compared to no calving pens in indoor systems; but are there studies comparing them with pasture systems? isn't calving on pasture very hygienic besides allowing for natural mother-calf-interaction?

We share the same opinion of the reviewer in these aspects. However, in general, the confinement allows a better control of these issues, mainly when we work with a large size farm. We rewrote the paragraph.

Lines 362-366 

L405. ). Despite this, only 30% of farms implemented quarantine measures when introducing animals. Did you show this data before?

We have changed the sentence: “ Despite this, only 38.5% of the respondent farmers (see Table S1)”

Lines 409-410

L410: Yes, but is this really comparable? Yours, after all, is an island and Belgium is in the middle of Europe...

We agree with the referee and we added: “The different kinds of production systems found in Azores, probably contribute to these differences.”

Lines 415-416

L418: Again, this is a very general thing to say; are there specific recommendations? what is the state of the art in this respect? 

We have rewritten the paragraph and include a reference.

Lines 423-425

L431: does that mean one could also argue, that nutrition (analyses, feed rations etc.) is less critical if yields are lower, as in MMS farms?

Yes it is. We thank the suggestion of the referee. We changed the term by “ in postpartum dairy cows”.

Line 438

L463-465: please look into behaviour change literature and implementation science (e.g. Michie et al 2011); you will learn that education is only one possible intervention besides many others; by itself it may not be very effective at all; research is needed that looks into drivers of farmer behaviour (e.g. adoption of measures) and identifies barriers and facilitators of change just because you know something, doesn't mean you change your behaviour; drivers of behaviour (change) can be categorized into capability, opportunity and motivation (again, check out Michie et al. 2011).

We are thankful with this comment from the reviewer and we restructured and rewritten the entire paragraph.

Lines 487-489

L469: I am missing a discussion of your methods, i.e. your sample, your survey items, the fact you used farmer perceptions rather than health data to rank health problems etc.

We thank the referee for this comment. We have added some sentences of discussion into this section. For example:  “The dairy industry is a business which provides incomes to farmers, and is the strongest motivation for farmers. The production efficiency and herd health management improvements, and new opportunities coming from social demands, including consumers perspectives on animal welfare and environmental impact, seem to be crucial keys for behavioral changes.”

Lines 489-493

We also added: “In fact, the greater part of the health information reported in this study came from farmers’ perception. A health data record is important to objectively evaluate the herd health and take appropriate decisions, and can contribute to a low degree of health program implementation. Nevertheless, farmer education is only a part of the equation to drive behavioral change.“

Lines 483-487

We also added in 2.2 Survey: “Our questionnaire was completed by a preliminary assessment of management practices occurring in Azorean farms.”

Lines 130-131

L474: I wouldn't say they 'should'; but probably they would benefit from it in terms of animal welfare, productivity, resource efficiency, sustainability etc.

It has been changed.

Line 513

L476-478:This is an aspect I find very important (if not to say central) and discussion on that is still quite weak: How could the MMS farms transform, in order to keep their tradition but still increasing efficiency, animal welfare etc.?

We agree with the comment of the referee and we included a paragraph to clarify this idea. However, the main constraint/factors regrading farmer´s perception for health problems were discussed.

Lines 503-508

Reviewer 2 Report

  • dear Author, thanks for your article:
  • L74-85 heat stress maybe not so relevant for the aim of investigation and there will be no acquisation in the rest of paper
  • Aims can be better framed, it's a bit general (that farm management has an high impact on animal health is not new, but to find the main influencing factors for your production system would be interesting)
  • especially in the results you should use anove or statistical analysis to put all your parameters in a model an get exacter results. So the results are very general.... 
  • Lameness seems really high for these production systems- why, and will it influence the other parameters. You should discuss that.
  • Which are your main conclusions, what should  be done and implemented first?
  • was it right idea to use milking system as the main splitting factor?

Author Response

First, we would like to thank again the reviewers for their thorough review of the manuscript, and for their constructive feedback and pertinent comments. We have highlighted in the manuscript the changes from Referee1 in blueyellow for Referee 2 and green for Referee 3.

We would like to thank the revision from the second referee. She/he helped us to improve substantially the manuscript with her/his suggestions, which have been included throughout the manuscript and highlighted in yellow.

  L74-85 heat stress maybe not so relevant for the aim of investigation and there will be no acquisation in the rest of paper

The entire paragraph was deleted

  Aims can be better framed, it's a bit general (that farm management has an high impact on animal health is not new, but to find the main influencing factors for your production system would be interesting)

We specified the objectives of the study in the simple summary and in the end of the introduction.

Lines 102-105 highlighted in blue, because, ref 1 also ask a clarification.

  especially in the results you should use anove or statistical analysis to put all your parameters in a model an get exacter results.

 We added “A non-normal distribution of all continuous variables, including all five-point scales from categories of intensity/prevalence, was confirmed using the Shapiro–Wilk test. Therefore, a non-parametric one–way ANOVA model, followed by Van der Waerden post hoc analysis to test significance, was used (Haiko Luepsen, 2018).”

Lines 154-157

  Lameness seems really high for these production systems- why, and will it influence the other parameters. You should discuss that.

We do completely agree with the comment from the reviewer, so, we included a detailed paragraph of this discussion.

Lines 464-469

  Which are your main conclusions, what should  be done and implemented first?

The first main conclusion is that there is a clear difference in health management between the two types of farms, due to nature of the highest and quicky transition of FMP. Moreover, the implementation of preventive and structured control programs, assessed by professional advisors and enhancing health, welfare efficiency and profitability is essential in dairy cattle farms. Both are the real solution for both types of production system.

We rewrote the conclusion section to enhance clarity.

Lines 510-512 and 514-516

  was it right idea to use milking system as the main splitting factor?

The authors think that this is as independent objective variable, linked directly to the characteristic of the farm splatted between traditional versus intensified. We have added this explanation in the M&M section

Lines 122-123 highlighted in blue, because, ref 1 suggested this addition as well.

Reviewer 3 Report

Dear Dr. Simões,

Your manuscript represents a highly relevant and extensive study for a unique setting of cattle production in the Azores Islands. 

The manuscript is well written, providing the reader with an excellent insight in the management of grazing cattle in Azores.

Minor comments:

In lines 184-186, the authors describe 2 variables (Refrigerated milk bulk tanks and inclusion into an official animal welfare program), with their respective p-values, however, when the percentages are described for FMP and MMS, only one percentage is described, so it is not clear to which variable does it belong or is a percentage is missing. In addition, it looks like the p-value of p<0.05 is repeated or missing. These sentences need clarification.

L254: I think the authors wanted to say unifeed instead of "unified"

L275: The authors describe "a lameness incidence peak". When does this peak take place? How often? Please clarify in the manuscript.

Table 4: Vaccination: there is a mistake in the sample size for vaccination against Clostridial diseases in MMS farms (should 2 instead of 1)

Table 4 and table s2 and discussion: It looks like vaccine utilisation it is quite low in the farms studied, in general terms, compared to cattle production in developed countries, however the authors fail to discuss this fact in the manuscript. It would be interesting to discuss why those lower vaccination percentages were observed in grazing cattle in Azores.

Author Response

First, we would like to thank again the reviewers for their thorough review of the manuscript, and for their constructive feedback and pertinent comments. We have highlighted in the manuscript the changes from Referee1 in blueyellow for Referee 2 and green for Referee 3.

Dear Dr. Simões,

Your manuscript represents a highly relevant and extensive study for a unique setting of cattle production in the Azores Islands. 

The manuscript is well written, providing the reader with an excellent insight in the management of grazing cattle in Azores.

Minor comments:

In lines 184-186, the authors describe 2 variables (Refrigerated milk bulk tanks and inclusion into an official animal welfare program), with their respective p-values, however, when the percentages are described for FMP and MMS, only one percentage is described, so it is not clear to which variable does it belong or is a percentage is missing. In addition, it looks like the p-value of p<0.05 is repeated or missing. These sentences need clarification.

We thank the comment from the referee and we rewrote the sentence and include data to enhance clarity.

Lines 174-176

L254: I think the authors wanted to say unifeed instead of "unified"

Corrected.

Line 250

L275: The authors describe "a lameness incidence peak". When does this peak take place? How often? Please clarify in the manuscript.

We included the specification: “…was only performed as treatment after the detection of lameness …” in order to enhance clarity as the reviewer suggested.

Line 274

Table 4: Vaccination: there is a mistake in the sample size for vaccination against Clostridial diseases in MMS farms (should 2 instead of 1).

Corrected in table 4.

Table 4 and table s2 and discussion: It looks like vaccine utilisation it is quite low in the farms studied, in general terms, compared to cattle production in developed countries, however the authors fail to discuss this fact in the manuscript. It would be interesting to discuss why those lower vaccination percentages were observed in grazing cattle in Azores.

We agree with the reviewer and we included a specifical sentence discussing this idea: “The low vaccination rate observed in our study was probably influenced by the immediate cost of vaccination, lack of immediate apparent health improvement, previous experience of failure to control the disease in the farm during two-years period and educational fail of farmers to implement vaccinal protocols.”

Lines 474-476

Round 2

Reviewer 2 Report

no comments